# Survival Is Skin Deep: Toughness of the Outer Cactus Stem with Insights for Technical Envelopes

**DOI:** 10.3390/biomimetics10080487

**Published:** 2025-07-23

**Authors:** Patricia Soffiatti, Natália O. Bonfante, Maria Clara L. Jaculiski, Nick P. Rowe

**Affiliations:** 1Botany Post Graduate Programme, Department of Botany, Federal University of Parana (UFPR), Curitiba 81531-990, Brazil; nataliabonfante@gmail.com (N.O.B.); mariamclj@gmail.com (M.C.L.J.); 2AMAP, University of Montpellier, CIRAD, CNRS, INRAE, IRD, 34398 Montpellier, France

**Keywords:** anatomy, coastal dry forests, cactus skin, extreme environments, growth forms, toughness, hypodermis

## Abstract

Cacti are of interest for new bio-inspired technologies because of their remarkable adaptations to extreme environments. Recently, they have inspired functional designs from nano fibres to optimised buildings and architectures. We investigate the diversity of cactus skin properties in terms of toughness and resistance to cutting damage. Cacti are well known for their extreme adaptations to harsh environments, with soft, fleshy stems that expand and contract with water uptake and storage. This functioning is made possible by an extendable outer envelope (skin) and a fluted 3-dimensional structure of the stem. We explore the mechanical toughness and underlying structural organisation of the cactus skin in four species of cactus showing different growth forms. The toughness properties of the cactus skin is only one part of a multi-functional structure for surviving in extreme environments. The study suggests that survival involves a relatively “light” investment of tough materials in the outer envelope instead of a rigid “defensive” layer. This is capable of elastic deformation and enables water storage in challenging, arid environments. The main purpose of this article is to demonstrate the diversity of skin toughness and underlying structures in the biological world as providing potential new designs for technical envelopes.

## 1. Introduction

Combating global change requires conservation of biodiversity and a maintenance of the world’s biomes and ecosystems. There is also a growing need for new, safe and sustainable technologies that are efficient in terms of energy consumption and material cost while faced with a changing environment and living conditions. Coherent with these two general needs is a rapidly growing interest in bio-inspired concepts from living organisms and incorporating these into new technologies. While many new designs are based on animals, innovative technologies based on plants are also providing technical breakthroughs via entirely different life processes. Phenomena linked to climate change, such as high winds, high temperatures, flooding, drought, herbivory, and pests, represent extreme conditions that plants have to survive. In this article we investigate the structure and mechanical toughness of the outer envelope of the cactus stem known as the cactus “skin”. This outer layer is crucial for protecting the soft hydrated tissue inside and surviving extreme environments and damage from pests.

Cactus skin is a protective barrier against severe biotic and abiotic factors in the extreme conditions in which these plants usually grow. Herbivory in cacti has not been studied extensively; however, reports demonstrate that insects are the most common herbivores, with reports of termites, beetles, and ants [1]. Many insect larvae rupture the skin and invade the internal parenchyma, feeding on the soft tissues [2]. Also, vertebrates, including birds, rodents, squirrels and rabbits, have been recorded to cause damage to cacti, as well as bacteria, fungi and parasitic plants [2,3,4,5,6]. Abiotic factors caused by solar radiation or frost have also been reported as potentially damaging for cacti [7,8,9,10].

Today, cacti are a rich source of models for bio-inspired technologies across a vast range of scales, from nano structures to optimised buildings and architecture. Compared to many other plants, cactus organisation is strongly adapted to extreme environmental conditions and thus offers attractive technologies for environmental extremes. Some examples of recent cactus-based innovations include multi-material composites mimicking plant movements as a new kind of actuator requiring no energy input [11,12]; optimising fracture behaviour of materials based on cactus tissue organisation [13]; water harvesting technologies in arid environments [14]; mechanical innovations for optimising crushing mechanics for automobile safety [15]; innovative cactus-inspired buildings [16]; functionally adaptive facades for optimising living conditions in buildings [17]; and fabricating optimised electromagnetic wave adsorption surfaces mimicking the nano structure of the cactus surface [18].

These recent designs all depend on biological knowledge of external and internal organisation and the properties and functionalities of the component parts of the model organism. Today, there is an increasing interest in bio-inspired approaches for new technologies. It is becoming clear that the diversity of informative models [19] has much to contribute to technical innovation.

We investigate the toughness properties of the cactus skin in four cactus species from the seasonally dry restinga forest in southern Brazil, representing part of the Atlantic Forest biome. Brazil is home to six different biomes, ranging from the Amazon and Atlantic Forest with tropical humid rainforests to the seasonally dry Cerrado and Caatinga. As well as representing some of the world’s most important environmental heritage sites, these biomes offer a great diversity of plant form and function for potential bio-inspired mechanisms [20] adapted for extreme environments such as cacti [21]. Coastal plain forests, or restinga forests, are seasonal heterogeneous coastal forest formations found in the south and southeast of Brazil. They are characterised by poor and dry sandy soils but comprise rich ecosystems [22]. Although, restinga formations are home to very rich floras and faunas they are highly threatened by urban degradation and pollution [23].

In cactus stems, the skin is the first line of defence against arid conditions. The “simple” organisation belies a high level of multifunctionality that has a potential for technical applications [12,21]. It must be mechanically resistant to form a safe and reliable envelope to enclose soft water-storing tissues. It must also allow light to reach the photosynthesising tissue inside the stem, and at the same time, it must limit evaporation of water from the inside to the outside. The surface of cactus skin includes special regions called areoles, which give rise to the spines, which are adapted for defence, shading of the cactus skin from the sun and even climbing. Spine anchorage of climbing cacti must be mechanically reliable and, in fact, shows “fail-safe” and benign failure properties when under mechanical stress [21,24]. The cactus skin also shows properties linked to swelling and de-swelling of the stem that are necessary for water storage under conditions of drought [11]. Finally, some species show “self-repair” of the skin tissue following the development of water-searching roots that emerge from inside the stem and which puncture the skin [24]. Self-repair has also been observed following experimental cuts to the cactus skin and underlying cortex [25].

Cuticle is the ultimate barrier between the plant body and the environment, deposited on the outer cell walls of the plant epidermis. The cuticle performs numerous functions, from mechanical protection against biotic and abiotic factors to controlling gas exchange and water loss, protecting the plant body against UVA-B radiation, and controlling the development and growth of plant organs together with the epidermis [26,27]. This multifunctional component combines the properties of extensibility and rigidity and facilitates changes in morphology during organ development [28]. In Cactaceae, together with the epidermis and hypodermis, it forms the “skin”, a complex and efficient barrier that protects and allows leafless stems to be fully photosynthetic while accommodating changes in volume of the succulent stems and preventing mechanical failure and the effects of biotic and abiotic stresses [29,30].

Due to its importance in the quality of crops, many mechanical studies of cuticles focus on tomato fruits, with tensile tests of isolated cuticle samples [26,27,31]. Other studies have addressed the mechanics of leaf cuticles in tension, e.g., [32].

A few studies have explored the biomechanics of the plant “skin”—the term which includes the cuticle, epidermis, and hypodermis. In apple fruits, one study [33] focused on the roles of epidermal and hypodermal cells and their mechanical properties subject to russet disease. This demonstrated the importance of cell size and homogeneity under high strain fracture and, therefore, resistance to the disease. Some authors studied the mechanics of skin under tension, but in relation to the properties of branch junctions [34], identifying that the tensile modulus of dermal tissues increases when a periderm replaces the skin. One study [35] investigated the stem tissue mechanics of three cactus species, including the skin, in order to understand branching point stability and aimed to contribute to the development of new concepts for fibre-reinforced composites. Finally, another study [12] tested directly the mechanical properties of the skin to elucidate how much it could expand or shrink in order to accommodate turgor or wilt when the stems were subjected to extreme hydric conditions.

In this paper we compare the mechanical toughness of the skin of four species of cactus by carrying out razor-cutting tests on samples of skin excised from the cactus stem. The approach was chosen to provide values of toughness Jm^−2^ in terms of the energy required to propagate a fracture per unit area of material thickness. Specifically, we aimed to explore the following questions:

(1) Different growth forms of plants, such as differences between shrubs, trees and climbers, can show very different mechanical strategies. Do diverse growth forms of cacti have different properties of toughness?

(2) Plant stems often show asymmetrical mechanical properties across their stem shapes and geometries; for example, many plant stems fail more readily under compression than tension in bending. Is skin toughness the same in vertical and horizontal orientations of the stem?

(3) Mechanical properties in plant stems often change from young to old stages of growth; do cactus skins show changes in toughness from young apical to older, more basal stages of growth?

(4) We explored the structural organisation of cactus skins. We aimed to discover whether the different cacti had the same basic organisation or whether across diverse species skin toughness was constructed in different ways.

(5) Finally, we briefly discuss how the mechanical and structural properties of cactus skin from extreme environments can contribute to design principles for technical envelope resistance.

## 2. Materials and Methods

### 2.1. Location and Material

We selected three growth habits among diverse species of cactus from the Atlantic Forest biome of southern Brazil. Plant stems were collected from the restinga coastal lowland dry forests near the town of Armação dos Buzios (22°44′49″ S, 41°52′54″ W, 174 km from the city of Rio de Janeiro), Rio de Janeiro State, Brazil. The area includes a mosaic of vegetation types with numerous species of cactus with a climate that is typically warm and seasonally dry (Figure 1). The so-called restinga forests are lowland dry forests composed of herbs, trees and shrubs. *P. arrabidae* (Figure 1A) and *P. ulei* (Figure 1B) are co-occurring columnar species of the restinga coastal grasslands and shrubby low-stature formations. They are also present, growing in exposed rocky formations along the coast of Armação dos Buzios [36]. *C. fernambucensis* is a scrambling-creeping species (Figure 1C) which typically occurs in herbaceous restinga formations, the so-called coastal grasslands [37].

*Selenocereus setaceus* is a climbing species that is common in the region and climbs on trees and shrubs. All of the species showed visible evidence of wounding as open holes and fissures in the cactus skin (Figure 2A–D), with some wounds associated with the loss of entire branches. The observations highlight the importance of cactus skin for all kinds of growth forms observed.

### 2.2. Specimen Preparation

Cutting tests providing measures of “skin” toughness were carried out on the outermost layer of healthy stem segments for each species. Three stem segments were harvested from three different individuals from the same habitat and location. The samples for all species were strictly limited to minimise any effects of the sampling on the species survival in this area, some of which are endemic to the region, and to limit any damage to the fragile environment. Segments up to half a metre in length were carefully pruned from healthy individuals using a saw and wrapped in newspaper and kept in the shade prior to testing.

Just before testing, strips of cactus skin were dissected from the stem surface, kept in moist tissue paper and wrapped in plastic. Segments were cut from the upper (young) part of the stem, the middle region of the segment and the older basal part of the stem segment. At each level, two strips of skin approximately 10 × 35 mm were cut away with a fresh scalpel in a longitudinal and transverse orientation relative to the stem’s main axis. The thickness of each strip was pared back to a thickness of 0.9–1.2 mm with a sharp knife, without damaging the integrity of the tissues forming the skin. This included the intact cuticle and hypodermis (the skin—the main interest of the study) as well as a layer of very soft internal cortical tissue [11]. Prior to each test the width of the skin sample (i.e., the length of the cut) and the thickness of the sample were measured with a digital calliper to 0.01 mm precision.

### 2.3. Cutting Test

Strips of cactus skin were then submitted to a cutting test using a modification of the single inclined razor blade test [38] using a portable Instron testing device (In-Spec2200, Instron Corporation, Norwood, MA, USA) (Figure 3). Each strip of skin was placed on a platform beneath the crosshead of the Instron. This comprised two horizontal metal plates separated by a gap of 1.3 mm. The strip of skin was firmly anchored in place with two strips of Scotch tape against the basal plate at each side of the gap.

For each test a single-edge razor blade (Gem Personna 3-Facet 0.23 mm) was clamped in the crosshead grip at an angle of 45° to the surface of the cactus skin (Figure 3). Blades were removed from the aluminium holder, cleaned of grease with 100% alcohol and conditioned by cutting across a 20 cm length of a photocopying paper sheet of paper 5 times to standardise the sharpness of the blade. Pilot studies found that using fresh blades significantly modified the cutting properties of initial specimens of a series if fresh, new blades were used. Conditioning the blade in this way meant that blade sharpness remained constant for up to 10 cutting experiments.

During each test, the blade was lowered towards the specimen at a rate of 0.32 mm per second until the strip of cactus was cut through completely. (Figure 3) The exact point at which the blade started cutting the skin (start point) and at which the blade finished cutting through the skin at the end of the test (end point) was observed with a magnifying glass and later marked on the load deflection graph (Figure 3C). During this first cutting pass the Instron device recorded the load (N) and displacement (mm) at 50 measurements per second. Following this initial cut and following the procedure in [38], the blade was cleaned in 70° alcohol with a cotton swab and returned to the start position. A second pass was then carried out measuring the force resulting from the friction of the blade against the sides of the skin specimen. The device interface plotted this second curve alongside the initial cutting plot (Figure 3C).

Following each test, the computer interface calculated the fracture toughness of the skin after (1) the start and end points had been marked on the graphs of the cutting and friction curves, (2) the actual cut distance measured with callipers, and (3) the thickness of the sample (skin plus cortical tissue) measured with callipers. We calculated the toughness of the sample following each test. However, because this included a layer of soft cortical tissue in the sample preparation, we adjusted values of toughness by calculating toughness based on the thickness of the cactus skin measured via anatomical sections (see below) and neglecting any minimal contribution from the soft cortical tissue.

### 2.4. Fracture Toughness of Skin

The equation describing the toughness (*R*) of the cactus skin is the following:*R* = J m^−2^

where J = (total energy of cutting, surface beneath curve of pass 1 − friction energy, surface beneath during pass 2; areas beneath both curves were calculated for the areas between the observed start and end points observed during testing).

Where m^2^ = area of skin cut (length of cut (measured by calliper before test) × thickness of skin measured from anatomical micrographs).

Interpreting cutting tests performed on heterogeneous biological structures is complex. This is especially the case in terms of differentiating the energy due to elastic bending forces of the blade against the specimen, which are not directly linked to propagating a fracture. For this reason, we elected to pinpoint visually the exact point when the blade started to actually cut the sample as opposed to merely bending it at the beginning of the test (Figure 3). We also chose as an endpoint the visual emergence of the blade. Though not ideal, since there was undoubtedly some remaining elastic storage in the force/displacement curve, this approach eliminated a large part of the solely elastic energy increase at the beginning of the test and the release of energy at the end of the test. Using this approach, we used the distance between the start and end points of cutting for deriving the mean cutting energy while using the cut length measured on the actual specimen to calculate the cross-sectional surface cut to calculate toughness. For most of the specimens treated this way, the start-end point length approximated the sample measured cut more closely. Finally, we cannot completely differentiate between the build-ups and releases of elastic energy and increases–decreases in resistance due to patches of more resistant structures in the samples, such as thick-walled, vascular strands and surface deposits such as waxes.

### 2.5. Skin Structure

Following each cutting test, specimens of the cactus skins were stored in 40% alcohol and prepared for sectioning on a vibratome (Campden 5100 mz, Campden Instruments, Loughborough, UK). Specimens approximately 5 × 5 mm in length and width were mounted on the vibratome platform with cyanoacrylate adhesive. The specimen was aligned in such a way that sections were cut close to and exactly parallel to the original cut made during the toughness test. Sections were then stained in Safranin and Astra Blue (1:9), mounted temporarily in distilled water, and observed with a compound light microscope where high-resolution images were made of the skin cross-sections. Measurements of the skin thickness and cell wall measurements of the key tissues were made with the image analysis Optimas, where simple linear measurements were made manually using the interface cursor on directly on high-resolution images of the anatomical organisation.

All quantitative data were analysed using the software PAST4: Paleontological Statistics Software Package for Education and Data Analysis [39].

## 3. Results

### 3.1. Damage of Cactus Skin in the Natural Environment

Field observations of the cacti in their natural habitats showed evidence of skin damage in all four species. Damage was present as circular to oval wounds on the surface of the cactus ribs (Figure 2) ranging from several mm to over a cm in diameter and occurred in relatively young segments close to the stem apex. Some of the damage appeared along the edge of the cactus ribs (Figure 2C), and in the self-supporting and creeping-upright species, it was present despite layers of defensive spines. Both marginal damage and surface puncturing were present in the climbing species in which protective spine cover is not as well expressed as in the other life forms, since spines are modified for climbing (Figure 2D).

### 3.2. Toughness of Cactus Growth Forms

The energy of cutting (J) (the energy required to propagate a fracture without considering skin thickness) varied significantly between different species (KW; *p* < 0.01) (Figure 4A). Both self-supporting species, *P. arrabidae* (0.009 ± 0.007 J) and *P. ulei* (0.009 ± 0.003 J), showed moderately higher cutting energies compared to the creeping-upright *C. fernambucensis* (0.006 ± 0.002 J) and the climbing species, *S. setaceus* (0.003 ± 0.002 J). The latter showed consistently lower values of energy of cutting than the other growth forms (Figure 4A, Table 1).

Values of skin toughness (Jm^−2^) calculated by considering skin thickness mirrored values of energy of cutting (Figure 4B) with significant differences between species (KW; *p* < 0.01). Both self-supporting species showed higher toughness values (*P. arrabidae* 5346.82 ± 2940.29 Jm^−2^ and *P. ulei* 4304.94 ± 1357.18 Jm^−2^) than the creeping-upright *C. fernambucensis* (3912.97 ± 1175.34 Jm^−2^) and the climbing *S. setaceus* (1642.52 ± 695.32 Jm^−2^).

Cutting tests showed surprisingly little difference between tests orientated in the vertical (top to bottom) direction of the stem compared with tests in the transverse direction. None of the comparisons showed statistical significance (Table 2).

### 3.3. Toughness and Developmental Age

A plot of fracture toughness at variable distances from the apex for each measured segment showed no clear overall tendance when all four species were analysed together as shown by a very low overall R^2^ value (Figure 5A). When analysed as distinct species, three out of the four species showed significant slopes but weakly supported R^2^ values for increases in toughness with increasing distance from the apex. In these cases, aproximately 25% of stem toughness variation is explained by stem maturity for *P. arrabidae* (R^2^ = 0.282; *p* < 0.01); *P. ulei* (R^2^ = 0.2619; *p* < 0.01) and *S. setaceus* (R^2^ = 0.2303; *p* < 0.01). However, the trend seen in the creeping-upright species *C. fernambucensis*, presented only an extremely low R^2^ value, with no statistical significance (R^2^ = 0.0002; *p* > 0.05). Though necessarily based on few samples, the degree of increase in toughness varied significantly between species with the largest difference of a 7-fold increase in the large-bodied upright *P. arrabidae* from *c.* 2000–14,000 Jm^−2^ and the smallest increase seen in the climber *S. setaceus* with an approximately 2-fold increase from *c.* 1000 to *c.* 3000 Jm^−^^2^ (Figure 5B).

### 3.4. Structural Organisation of the Cactus Skin

The general anatomical organisation of the cactus skin was similar in all four species, forming a relatively dense layer compared to the soft cortical tissue it protects (Figure 6A). All four species comprise an outer waxy cuticle; this is pink to red-stained in the micrographs (Figure 6A–E). Immediately inside the cuticle is a single layer of thin-walled epidermal cells, which vary in size and shape between species but all show a similar low density of wall material (Figure 6B–E).

The main anatomical feature distinguishing all four species of cactus is the nature of the hypodermal layer immediately below the epidermis. These varied from three to four cell layers of irregular primary thick-walled cells with very small lumens and included prismatic crystals in the self-supporting *P. ulei* (Figure 6D); dense and less fused primary thick-walled tissue in *P. arrabidae* (Figure 6C); one- to three-cell-thick layer of rounded to oval cells with thick primary walls in *C. fernambucensis* (Figure 6B); and a two- to three-cell-thick layer of relatively thinner primary walled cells in *S. setaceus*. Overall, the hypodermis of both self-supporting shrubs showed thicker cell walls with little or no intercellular spaces and little differentiation between neighbouring cell walls and small lumens; the creeping-upright and climbing species showed less dense and less fused cells, with some intercellular spaces that retained ellipsoid cell outlines and thinner cell walls.

The overall dimensions of the skin and its component parts were highly variable between species and growth forms (Figure 7). Cuticle thickness across all four species differed between the species, varying from 8.06 µm in *S. setaceus* to 11.30 µm thick in *P. ulei*. The climbing species, *S. setaceus*, had a thinner cuticle compared to the other three species, while *C. fernambucensis* had intermediate values to *P. arrabidae* and *P. ulei* (Table 2; Figure 7A).

Hypodermal thickness differed between cactus species: *P. ulei* (0.13 mm) and *S. setaceus* (0.12 mm) had relatively thicker hypoderms representing *c.* 80% of the skin cross-section compared to *C. fernambucensis* (0.08 mm) and *P. arrabidae* (0.09 mm), representing *c.* 70% of the skin cross-section (Figure 7B, Table 2).

Overall skin thickness also varied between these same pairs of species: a thick-skinned group included the other self-supporting shrub, *P. ulei*, and the climber *S. setaceus*, 0.17 mm and 0.15 mm, respectively (Table 2; Figure 7C); and a “thin-skinned group” of the creeping-upright *C. fernambucensis* and the self-supporting *P. arrabidae*, both with an overall skin thickness of 0.12 mm.

## 4. Discussion

### 4.1. Skin Damage

Field observations indicated that skin damage was widespread and common in all four species of cactus and all growth forms. This suggests that damage, risk of water loss and further negative effects, such as necrosis and rotting of the stem—all of which were observed under field conditions—are real risks in addition to environmental extremes of temperature and humidity.

Our observations also indicated that spine cover does not necessarily defend the cactus stem completely from attack and damage [40]. All three growth forms with dense covers of protective spines showed evidence of herbivore damage, even in relatively young stems, suggesting that wounding of the skin is an ever-present risk in this environment despite defensive spines. Maintaining an elastically dynamic, living outer envelope that can adapt to water uptake (swelling) and water loss (shrinking) is liable to risks of damage in the environment, particularly by mammal, bird and insect actions.

### 4.2. Skin Toughness and Growth Form Diversity

Skin toughness and organisation varied with the type of cactus growth form. Self-supporting, shrub-like cacti of *Pilosocereus arrabidae* and *Pilosocereus ulei* showed approximately double the stem toughness that was seen in the climbing cactus *Selenicereus setaceus*. Interestingly, the physically “intermediate” creeping-upright growth form of *Cereus fernambucensis* was also mechanically intermediate in terms of skin toughness. This general comparison suggests that the skin of the cactus stem is mechanically diverse. Different growth forms appear to have evolved somewhat different skin properties, which are possibly linked to their differing exposures to the environment. Both self-supporting shrubs and the creeping-upright growth forms growing in exposed rocky habitats show high levels of skin toughness, whereas the climbing species often growing in more sheltered habitats on and among trees develop less skin toughness.

Apart from defence against abiotic and biotic risks, different growth forms also vary in terms of life-history mechanisms, which likely influence the need and expenditure for high levels of skin toughness. The shrub-like and partly creeping species develop via slow growth, producing robust stems and branches, high levels of costly defensive spines and long-lived, self-supporting individuals. The climbing species has a much lighter construction with slender stems and diminutive spines adapted for climbing and anchorage. It also has a so-called hemi-epiphytic life-history—older plants of the stem and branch system die back to allow younger apical stem parts to actively continue climbing growth [21,24]. The skin construction in cacti appears to be fine-tuned to different ecological and life-history strategies. Such biological principles can potentially provide new ideas for bio-inspired technological concepts, where a precise function is selected to closely and economically provide just the right function.

Despite the differences in toughness between species, toughness values of vertical and transverse orientations on the stem were surprisingly homogeneous, and this was true for all growth forms. It appears that skin strength and resistance to fracture are not optimised for strains within the skin resulting from bending forces due to lack of waterand low turgor pressures, where one might expect splitting of the stem longitudinally along the stem [11]. Similarly, we observed no particular patterns of vertical or horizontal stem damage in the natural conditions.

### 4.3. Skin Toughness and Developmental Age

Three out of the four species tested showed an increase in toughness with position or age along the stem. However, these trends showed only weak or non-significant R^2^ values, suggesting that other factors influence skin structure and toughness. This is perhaps not surprising because of the high multi-functionality of the outer envelopes of these plants with functions such as gas exchange, water conservation and permeability to light. Both upright species, *P. arrabidae* and *P. ulei*, showed relatively large changes in toughness where older, more basal parts of the stem have tougher skin than more apical areas. The increase in toughness was less marked in the climbing species *S. setaceus*, but no consistent increase was observed in the creeping-upright stems of *C. fernambucensis*. This suggests that the more robust upright stems invest more in the toughness of the stem in older basal parts of the plant compared with the shorter and creeping stems and the climbing stems of the other species.

### 4.4. Skin Properties and Structure

Our study demonstrated that all four species of cactus and different growth forms develop the same cuticle–epidermis–hypodermis organisation, which is a common feature in most Cactaceae [29,30,41,42] But there are differences between species and growth forms. The main differences concern (a) the structure of the hypodermis in terms of the thickness and degree of homogenisation of thickened cell walls and (b) the thickness of the skin and hypodermis.

The diversity of cactus skins observed provide insight into how elastically deformable envelopes are structured and vary among cacti in nature. The three toughest Cactus skins all show a hypodermal layer with significantly fused and thickened cell walls in the hypodermis. We suspect that this is the main criterion for conferring toughness to the cactus skin. All three produce significantly tougher outer envelopes than the climbing species *S. setaceus*, which produces the least amount of thickening of the cells of the hypodermis.

This variation of the mechanical structure of the outer skin in Cactaceae is possibly a widespread phenomenon following the early appearance of the thick-walled collenchyma cells of the hypodermis in the evolution of Cactaceae [29,43,44]. It has also been observed in epiphytic species of Rhipsalideae and Hylocereeae, where a reduction in hypodermis thickness and cell wall thickness is observed. These plants are more slender and lighter than self-supporting, more succulent counterparts [44,45,46]. Interestingly, other observers [47,48] have discussed how the shift from self-supporting terrestrial ancestors to the epiphytic form led to the loss of several features, acquiring a less robust structure. Such changes are concordant with modified outer envelopes as they thrive in less arid conditions.

Interestingly, hypodermis “density” and hypodermis thickness appear to be modulated together to provide toughness. For example, the two self-supporting cacti with tough skins show different mechanical organisations. The thicker hypodermal *P. ulei* has a marginally higher energy of cutting compared to the thinner hypodermal *P. arrabidae*. But when force values are normalised by skin thickness to provide values of skin toughness, *P. arrabidae* shows somewhat higher toughness than *P. ulei*. Nevertheless, there appear to be different constructional principles between the two shrub-like cacti skins to provide a similar level of toughness via either thick or thinner hypodermal layers. While this study concentrated on observing the diversity of skin types from field locations, further studies could first carry out more precise laboratory measurements at the cell or cell wall scale to determine more exactly the specific properties of the skin tissues, particularly that of the hypodermis.

### 4.5. Bio-Inspired Cactus Skin

As a well-adapted group of plants to harsh and dry conditions, cacti offer a wide variety of bioinspired designs based on overall morphology, spines, body structure and components [17,49,50,51,52,53,54,55,56,57,58,59]. The diversity of cactus skin types also offers a number of concepts for new kinds of artificial envelopes [17,51,53,57,60]. The structure and organisation of the cactus skin could be considered an improvised functional system with a number of trade-offs between contrasting requirements. The underlying requirement for the cactus stem in extreme environments is the ability to expand and contract according to water availability and for maximising water storage and the mechanical structure of the stem [29]. The need to retain an elastically stable outer envelope which allows expansion and contraction of the inner soft stem but which provides an effective barrier to attack is a challenge. Long-lived shrub-like cacti in exposed situations have a tougher skin than other forms that have a faster growth and shorter life span. The presence of different kinds of hypodermal tissue in the skins is likely adapted for these different kinds of growth form requirements. The notion of the robustness and longevity of a biological and technical system is of increasing interest for the transfer of biological concepts to a broad range of technological technologies [54,58,61]. The two degrees of toughness we observe in South American cacti offer some principles behind designing technical envelopes that are, for example, built to last for a longer rather than a shorter technical lifetime. Interestingly, the less robust and less permanent envelope we observed operates over a limited time and is then reabsorbed for renewed growth as a hemi-epiphytic life history or broken down in the environment. The ability to mimic growth processes on which biological envelopes depend is highly desirable for adaptive technical envelopes. For example, the Tensairity^®^ system mitigates damage to the technical envelope via a polyurethane foam, which mimics, the flexible thin-walled cells in the plant stem that fill and seal the lesions and spaces [62].

Single traits from animals and plants can provide wonderful bio-inspired technical ideas. However, many biological traits such as the cactus skin have combined functional requirements [17,51,53,57]. Cacti must control for water loss and gain to stay alive and so they require an expandable and contractable skin. This must not only limit water loss but also allow for gas exchange and the passage of light for photosynthesis. These combined requirements are not necessarily compatible with providing a mechanical defence against herbivory. So, selecting a new structural organisation from a plant such as a cactus for a technical envelope might not just be a case of choosing and transferring one biological trait into a technical one. Exploring the biology and life history around and beyond a single trait can uncover how a single trait is able to function in parallel with many other “design” requirements for surviving in extreme and changeable conditions.

## Figures and Tables

**Figure 1 biomimetics-10-00487-f001:**
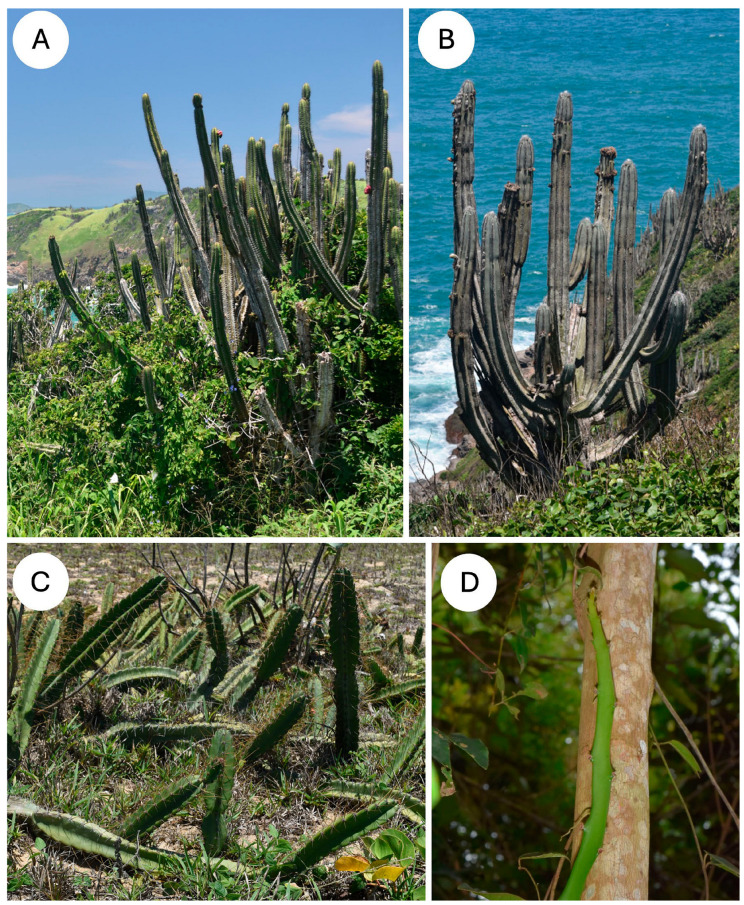
Growth forms and extreme environments of four cactus species (**A**) *Pilosocereus arrabidae,* shrub-like, self-supporting, branched growth form, often growing on exposed coastal locations with extremes in temperature and water availability. (**B**) *Pilosocereus ulei,* shrub-like, highly branched growth form with blue-green pigmented skin with surface wax growing in coastal locations in seasonally arid environments. (**C**) *Cereus fernambucensis,* upright and ground-creeping growth form growing in sandy coastal, exposed environments. (**D**) *Selenicereus setaceus*, slender, climbing and branched growth form common in sheltered Restinga forested locations and on isolated trees.

**Figure 2 biomimetics-10-00487-f002:**
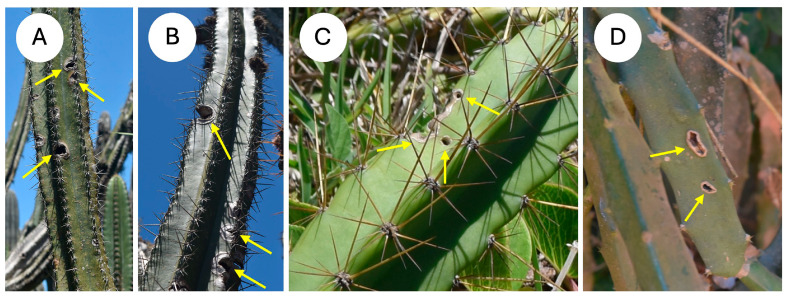
Skin damage under natural conditions (**A**) *Pilosocereus arrabidae,* open fissures on edges of stem margin (arrows) (**B**) *Pilosocereus ulei,* open fissures on edges of stem margin (arrows) (**C**) *Cereus fernambucensis*, rounded punctures on stem rib and lateral cut to rib edge (arrows). (**D**) *Selenicereus setaceus*, fissures in the mid-region of the cactus rib (arrows).

**Figure 3 biomimetics-10-00487-f003:**
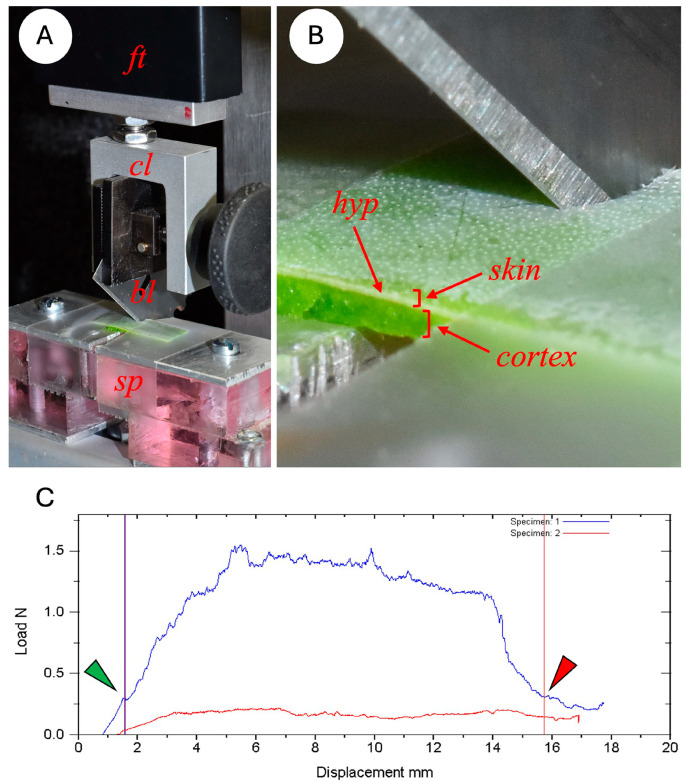
Experimental setup of the cutting test on Instron measuring device. (**A**) A 10 N force transducer (*ft*) is connected to the crosshead of the Instron; a clamp (cl) fixes the razor blade (bl) at an angle of 45° to the horizontally positioned cactus skin, (**B**) which spans a 1.3 mm gap in the metal support (sp). (**C**) Plot of the force-displacement curve of the skin cutting during the test. The energy of cutting (J) is the total area between the start and finish points (green and red arrows) of the first cutting pass (curve in blue) after subtracting the energy produced by friction in the second pass (curve in red, second cutting pass). Fracture toughness (Jm^−2^) is the energy of cutting divided by the area of the cut surface. The green arrow indicates the exact point of the beginning of the cut, which often occurs during the initial curve where elastic energy is building up in the specimen under load. The red arrow indicates the exact position where the blade has traversed the skin sample—this often occurs near to the end of the unloading of elastic energy of the specimen.

**Figure 4 biomimetics-10-00487-f004:**
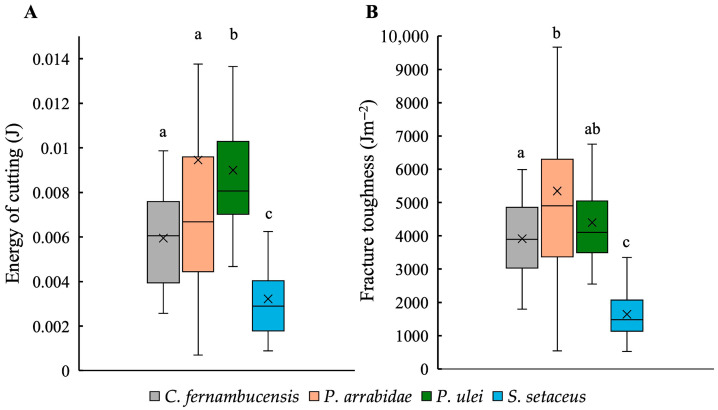
Boxplots showing the mechanical traits: (**A**) energy of cutting (J) and (**B**) fracture toughness (Jm^−2^) for the skin of the cacti species. (Box plots: inner lines, medians; X, mean values; boxes, 25th and 75th; whiskers, max and min values; excluding outliers). Small letters indicate significant differences at *p* < 0.01 (Kruskal–Wallis and post hoc Mann–Whitney) between medians.

**Figure 5 biomimetics-10-00487-f005:**
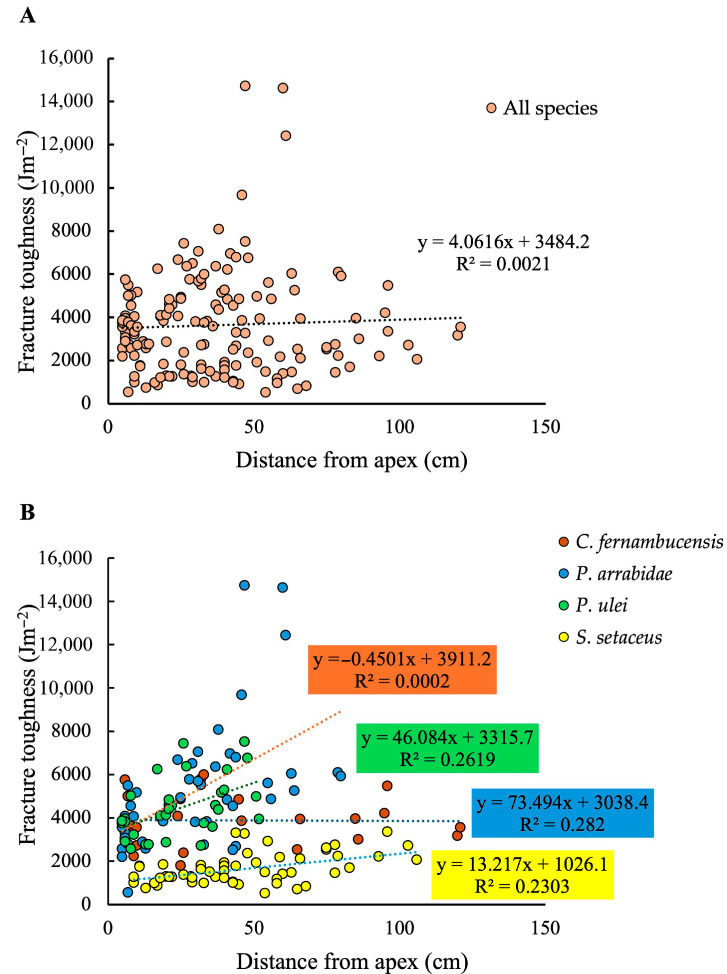
(**A**) Fracture toughness (Jm^−^^2^) plotted against distance from apex (cm) for all species. (**B**) Fracture toughness (J m^−2^) plotted against distance from apex (cm) for each species. There is a low correlation between toughness and stem maturity for all species (*P. arrabidae*, R^2^ = 0.282; *P. ulei*, R^2^ = 0.2619; *S. setaceus*, R^2^ = 0.2303), except for *C. fernambucensis*, which has a creeping growth form (R^2^ = 0.0002). When all species are considered together there is no trend or correlation between fracture toughness and stem maturity (**A**). When considered separately there is a correlation of aproximately 25% between the variables for the self-supporting and climbing species (**B**).

**Figure 6 biomimetics-10-00487-f006:**
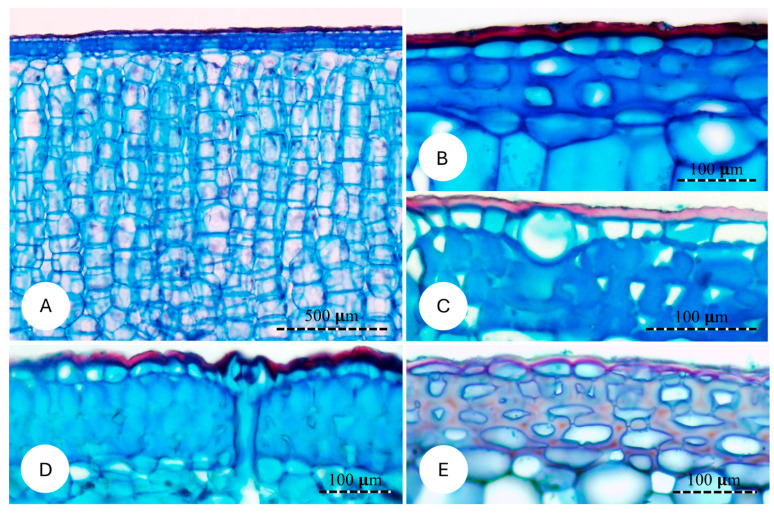
Cross sections of the skin of the studied species. (**A**,**B**) *Cereus fernambucensis;* (**A**) General view of the skin and its components. (**B**) High magnification showing the cuticle (external layer), uniseriate epidermis and three layers of hypodermis composed of cells with irregular thickened cell walls. (**C**) *Pilosocereus arrabidae*; hypodermal cells with highly irregular cell walls. (**D**) *Pilosocereus ulei*; irregular outer walls of epidermal cells; hypodermis is composed of 4–5 layers of cells with highly irregular cell wall thickening. (**E**) *Selenicereus setaceus*; hypodermis composed of cells with thinner cell walls.

**Figure 7 biomimetics-10-00487-f007:**
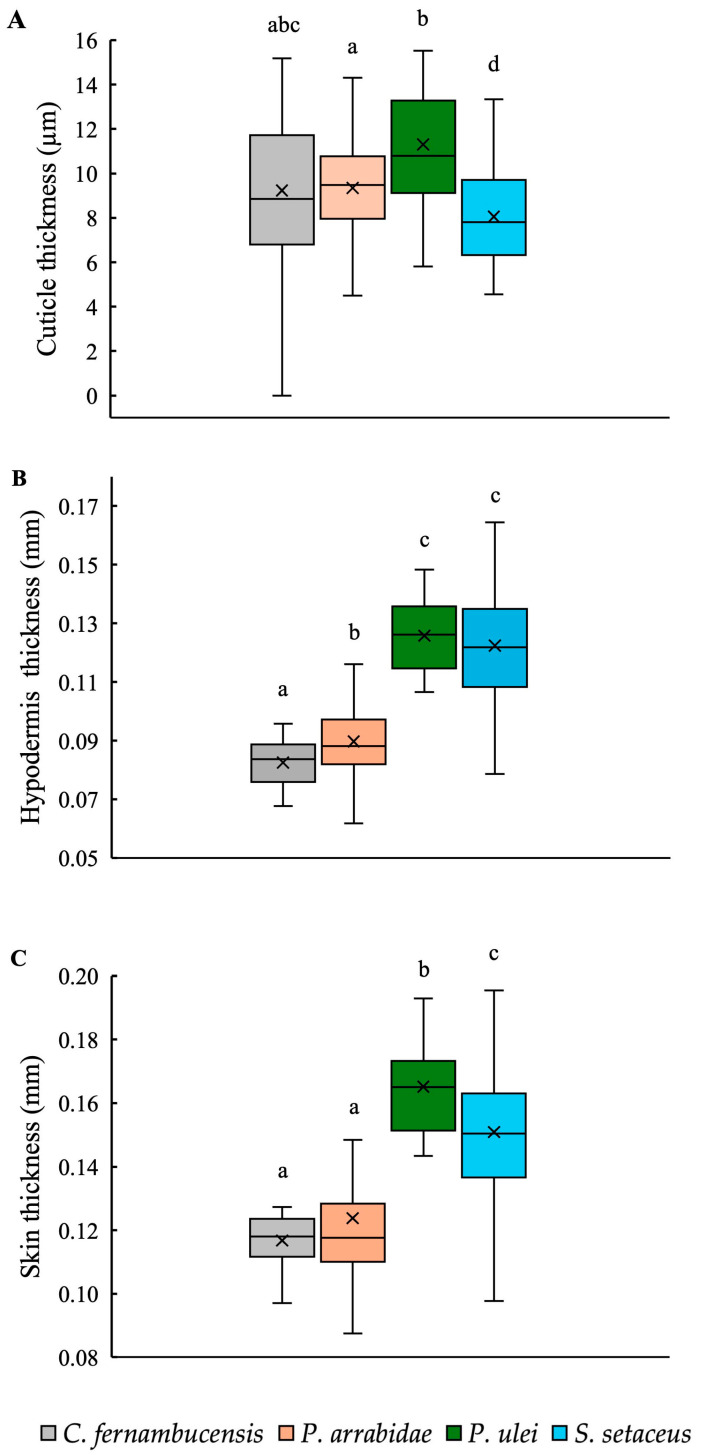
Box plots of (**A**) cuticle thickness (μm), (**B**) hypodermis thickness (μm), and (**C**) skin thickness (μm) of stems of the cacti species. (Box plots: inner lines, medians; X, mean values; boxes, 25th and 75th percentiles; whiskers, max and min values; excluding outliers). Small letters indicate significant differences at *p* < 0.01 (Kruskal–Wallis and post hoc Mann–Whitney) between medians.

**Table 1 biomimetics-10-00487-t001:** Mean values and standard deviation (between brackets) for anatomical and mechanical traits of stems of the species studied: *Cereus fernambucensis*, *Pilosocereus arrabidae*, *P. ulei*, and *Selenicereus setaceus*. There were statistically significant differences between the variables between species (Kruskal–Wallis and post hoc Mann–Whitney; different letters indicate significant differences at *p* < 0.01).

Traits/Species	*C. fernambucensis*	*P. arrabidae*	*P. ulei*	*S. setaceus*
Energy of cutting (J)	0.006 (±0.002) ^a^	0.009 (±0.007) ^a^	0.009 (±0.003) ^b^	0.003 (±0.002) ^c^
Toughness (Jm^−2^)	3912.97 (±1175.34) ^a^	5346.82 (±2940.29) ^b^	4398.71 (±1357.18) ^ab^	1642.52 (±695.32) ^c^
Skin thickness (mm)	0.12 (±0.008) ^a^	0.12 (±0.029) ^a^	0.17 (±0.018) ^b^	0.15 (±0.023) ^c^
Hypodermis thickness (mm)	0.08 (±0.008) ^a^	0.09 (±0.008) ^b^	0.13 (±0.016) ^c^	0.12 (±0.022) ^c^
Cuticle thickness (μm)	9.63 (±3.07) ^abc^	9.78 (±2.65) ^a^	11.30 (±3.29) ^b^	8.06 (±2.25) ^d^

**Table 2 biomimetics-10-00487-t002:** Mean values and standard deviation (between brackets) for the anatomical and mechanical traits of cross (CS) and longitudinal sections (LG) of stems of the species studied: *Cereus fernambucensis*, *Pilosocereus arrabidae*, *P. ulei* and *Selenicereus setaceus*. There was no statistically significant differences between the variables comparing cross and longitudinal sections for each species (Kruskal–Wallis and post hoc Mann–Whitney, *p* > 0.05).

Traits/Species	*C. fernambucensis*		*P. arrabidae*		*P. ulei*		*S. setaceus*	
Section	CS	LG	CS	LG	CS	LG	CS	LG
Skin thickness (mm)	0.12 (±0.009)	0.11 (±0.034)	0.13 (±0.025)	0.12 (±0.034)	0.17 (±0.017)	0.16 (±0.020)	0.15 (±0.022)	0.15 (±0.023)
Hypodermis thickness (mm)	0.08 (±0.008)	0.07 (±0.024)	0.09 (±0.015)	0.09 (±0.016)	0.13 (±0.016)	0.12 (±0.017)	0.12 (±0.021)	0.12 (±0.022)
Energy of cutting (J)	0.006 (±0.002)	0.006 (±0.002)	0.009 (±0.01)	0.010 (±0.011)	0.009 (±0.004)	0.009 (±0.002)	0.003 (±0.002)	0.004 (±0.002)
Toughness (Jm^−2^)	3788.24 (±3869.07)	3994.64 (±1119.05)	5009.86 (±2944.27)	5683.77 (±2965.75)	4304.94 (±1563.84)	4492.47 (±1155.30)	1678.76 (±663.64)	1606.28 (±736.47)

## Data Availability

Data will be made available when the manuscript is accepted for publication.

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
