# Peer review of "Survival Is Skin Deep: Toughness of the Outer Cactus Stem with Insights for Technical Envelopes"

_biomimetics, 2025, doi:10.3390/biomimetics10080487_

Round 1
Reviewer 1 Report
Comments and Suggestions for Authors
This manuscript is well written, clear, and concise. The experiments are well explained. I have a few comments that will improve clarity.
- When describing the hypodermis, it is mentioned that the cells are fused. This refers to the absence of intercellular spaces or to the fact that they are filled with the material that makes up the cell wall, in this case cellulose-hemicellulose, given that the hypodermis has primary walls. This point is important to clarify, since the characteristics of the hypodermis are of utmost importance in the discussion.
- Lines 428-434. Can you explain why these epidermal properties differ from those of other succulent trees or shrubs, where cracks form due to the increase in turgor during the rainy season? Could this be related to the late appearance of periderm in cacti?
- The reconstruction of ancestral stages in other epiphytic cacti and other growth forms supports this assertion. If so, this would mean that this is a general pattern in the family.
Author Response
- Comment 1: When describing the hypodermis, it is mentioned that the cells are fused. This refers to the absence of intercellular spaces or to the fact that they are filled with the material that makes up the cell wall, in this case cellulose-hemicellulose, given that the hypodermis has primary walls. This point is important to clarify, since the characteristics of the hypodermis are of utmost importance in the discussion.
- Response 1: Thank you for pointing this out. We have modified the text in the description of the hypodermis of the four species to “…Overall, the hypodermis of the two self-supporting shrubs showed thicker-walled and highly fused cell walls with little or no intercellular spaces and little differentiation between neighbouring cell walls, and small lumens; the creeping-upright and climbing species showed less dense and less fused cells, with some intercellular spaces that retained ellipsoid cell outlines and thinner cell walls." You can find the changes incorporated to the manuscript file in red. Lines 373-376.
- Comment 2: Lines 428-434. Can you explain why these epidermal properties differ from those of other succulent trees or shrubs, where cracks form due to the increase in turgor during the rainy season? Could this be related to the late appearance of periderm in cacti?
- Response 2: This is a little tricky to answer satisfactorily without further detailed studies that address the issue. We can say that in a previous study on the skin of Selencicereus setaceus, swelling tests caused splitting only of the periderm, in regions of the stem where the epidermis-hipodermis were no longer present. Periderm is a much more brittle tissue (composed of cells with lignified and suberized cells walls), which caused points of stress concentration which initiated the fracture. The hypodermis is a more flexible yet resistant structure, capable of considerable expansion and retraction which is fundamental in cacti, especially in the large bodied self-supporting species. Among succulents, cacti are the only group that present such a solid collenchyma hypodemis, which in plants in that the stem is the main photosynthetic organ and forms a large succulent cortex, is an important adaptive acquisition guaranteeing mechanical support and not shedding internal photosynthetic tissues. We suppose the occurrence of a well developed layer of collenchyma is related to the late appearance of periderm in the lineages in which the stems are the main photosythesizing organs.
- Comment 3: The reconstruction of ancestral stages in other epiphytic cacti and other growth forms supports this assertion. If so, this would mean that this is a general pattern in the family.
- Response 3:
So indeed this is a pattern in the family, as less succulent groups such as the epiphytic and climbing lineages have a less developed hypodermis, which can be related to less succulence and/or a lighter or less robust structure. We have added a short phrase underlining this fact in the discussion and it nicely explains why a structurally similar organisation exists in these different growth forms but with some variations.
We have modified the following text in the discussion: This modulation of mechanical structure of the outer skin in Cactaceae is possibly a widespread phenomenon following the early appearance of the thick-walled collenchyma cells of the hypodermis in the evolution of Cactaceae {Mauseth, 1997 #55; Martínez-Quezada, 2025 #39; Terrazas & Arias, 2003; Gibson & Nobel, 1986}. Lines 482-483.

Reviewer 2 Report
Comments and Suggestions for Authors
Some formulations are quite week i.e. “environmental conditions need not require a large”, “Different growth forms of plants” _ strongly advice to revise the entire manuscript
Form the abstract it is not clear which is the purpose of this paper
The authors mix the literature reviewe between plant and animal (insects) please be consistent. See line 43- 46 and some other
Teh authors are advised to keep consistent with the references style some time is used [] and some times (Marques et al. 2015).
The introduction requires more clarity as now it is difficult to understand its message
There is unclear how this work is different from literature and what can bring novel for research community and industry.
Images from Figure 1 are very fuzzy
“evidence of wounding as open holes and fissures of the cactus skin” please clearly highlight in that image were is that evidence claimed
“samples for all species were strictly limited to minimize” these sample align with any standards ?
Figure 3 is poor quality
Not clear what is the description for figure 3b
“Following each test, the computer interface calculated fracture toughness” can this be validated also numerically ?
“image analysis Optimas” the process of measurement should be described in details
Better quality for figure 4 is required
In figure 4 it is unclear the standard deviation, because above orange data are some extra points – the same apply for figure 7
A section of conclusion is required
Comments on the Quality of English LanguageThe English could be improved to more clearly express the research.
Author Response
-Comment 1: Some formulations are quite week i.e. “environmental conditions need not require a large”, “Different growth forms of plants” _ strongly advice to revise the entire manuscript
-Response 1: We have re-worded the “…environmental conditions…” phrase to clarify the concept and added to the manuscript Abstract (in blue), as follows: "The study suggests that survival involves a relatively “light” investment of tough materials in the outer envelope instead of a rigid “defensive” layer. This is capable of elastic deformation and enables water storage in challenging, arid environments." Lines 18-20.
The term “different growth forms of plants” is common and a widely understood construction in the biological literature; we have modified the phrase slightly and added to the manuscript (in blue) to make the meaning more accessible to a biological and technical audience: "1) Different growth forms of plants, such as differences between shrubs trees and climbers, can show very different mechanical strategies. Do diverse growth forms of cacti have different properties of toughness? " Lines 139-140.
- Comment 2: Form the abstract it is not clear which is the purpose of this paper
- Response 2: We point out that the purpose of the paper is very clearly defined at the end of the introduction with 5 main aims for the reader to understand what the paper is about. To address the concern of the reviewer we have added an explanation to the Abstract (in blue), which encapsulates the purpose of the paper, and which reflects our initial invitation to participate in this special volume. We feel that tit is up to the handling editor to decide whether this addition to the abstract is necessary. “The main purpose of this article is to demonstrate the diversity of skin toughness and underlying structures in the biological world as providing potential new designs for technical envelopes.” Lines 20-21.
- Comment 3: The authors mix the literature reviewe between plant and animal (insects) please be consistent. See line 43- 46 and some other
-Response 3: With all due respect we understand the reviewer’s concern concerning how factors causing damage to cacti are reported here. However, the literature dealing with this aspect is indeed diverse. For example the Martinez-Avalos et al. 2007 reference reports that 1) a fungus, 2) a beetle and 3) a squirrel are responsible for different kinds of damage and mortality for the cactus concerned. We have simplified the list of different factors and agents causing damage and merely put all the relevant references at the end of the section. The paragraph has been restructured. See lines 39-46.
- Comment 4: Teh authors are advised to keep consistent with the references style some time is used [] and some times (Marques et al. 2015).
- Response 4: Thank you the "Marques et al" paper somehow jumped out of the Endnote scan in preparing the manuscript. This has been rectified. Line 90.
- Comment 5: The introduction requires more clarity as now it is difficult to understand its message
- Response 5: We appeal to the handling editor to assess the information of the Introduction. We feel that the material and the argumentation clearly express the background and aims of the study. If the reviewer could be more precise about exactly what parts of the introduction are unsatisfactory we would be more than happy to revise parts that are unclear.
- Comment 6: There is unclear how this work is different from literature and what can bring novel for research community and industry.
- Response 6: With all due respect we are not clear when the reviewer states that “… There is unclear how this work is different from literature…”. Could the handling editor please advise us about what the reviewer is perhaps meaning here?
Secondly we were invited as biologists to provide a novel biological theme for potential bio-inspired design applications. It is not possible for us at this early stage of biological presentation of biophysical parameters to project any detailed potential industrial blueprints or prototypes These can only be presented after further deliberation by biologists and technical experts. This Journal must decide if it is to invite biologists to provide detailed potential biological concepts for bio-inspired applications and advise its reviewers accordingly whether more detailed industrial applications are a requirement or not.
- Comment 7: Images from Figure 1 are very fuzzy
- Response 7: Our apologies but the conversion of the entire text and figure file to a single pdf meant that figures were relatively low resolution. New figures with a better resolution were inserted in the manuscript file.
- Comment 8:“evidence of wounding as open holes and fissures of the cactus skin” please clearly highlight in that image were is that evidence claimed
- Response 8: Thank you, we have superimposed arrows onto the Figures 2 to indicate the exact positions of the damage to the cactus skin.
- Comment 9: “samples for all species were strictly limited to minimize” these sample align with any standards ?
- Response 9: We have clarified this phrase further (in blue) to show that our sampling aimed to provide scientific information about these plants but at the same time did not destroy large numbers of individuals. We have integrated the following text to the manuscript: “The samples for all species were strictly limited to minimize any effects of the sampling on the species survival in this area”. Lines 178-179.
- Comment 10: Figure 3 is poor quality
- Response 10: See explanation for Comment 7.
- Comment 11: Not clear what is the description for figure 3b
- Response 11: Thank you for pointing this out, the description for figure 3B has been added to the manuscript, as follows (in blue):
" Figure 3. Experimental set up of cutting test on Instron measuring device. A) A 10N force transducer (ft) is connected to the cross-head of the Instron, a clamp (cl) fixes the razor blade (bl) at an angle of 45° to the horizontally positioned cactus skin, B) which spans a 1.6 mm gap in the metal support (sp). C).... " Line 213.
- Comment 12: “Following each test, the computer interface calculated fracture toughness” can this be validated also numerically ?
- Response 12: By this we mean that the skin toughness is calculated and validated numerically by the computer interface after the basic test parameters, length, force etc have been entered into the programme. We feel that the text here very clearly expresses this and with all respect we do not see a need to expand on this further.
- Comment 13: “image analysis Optimas” the process of measurement should be described in details
- Response 13: We have added a short phrase in the manuscript (in blue) to indicate that the measurements were made manually, as follows: "Measurements of the skin thickness and cell wall measurements of the key tissues were made with the image analysis Optimas where simple linear measurements were made manually using the interface cursor on directly on high resolution images of the anatomical organisation." Lines 282-284.
- Comment 14: Better quality for figure 4 is required
- Response 14: See explanation for Comment 7.
- Comment 15: In figure 4 it is unclear the standard deviation, because above orange data are some extra points – the same apply for figure 7
- Response 15: In Figures 4 and 7 we present the boxplots and non-parametric rank based comparisons of these small samples. Small samples were obligatory because of collecting permits and precautions against endangering the species in this environment. We therefore do not provide “parametric” mean based statistical parameters based on means, standard deviations based on normal distributions.
We do provide means and standard deviations in the tables as part of the data presentation and in addition to the non-parametric statistical comparisons. Of course the means and standard deviations are less likely to be really true with these sample numbers but give the reader a further feel for the data in addition to the medians and data presented in the box plots.
- Comment 16: A section of conclusion is required
- Response 16: To our understanding the Journal guidelines do not insist on a separate conclusion section. For this short paper we think that a closing paragraph in the discussion is probably adequate for covering the final general conclusion of the work. If the handling editor thinks that a further conclusion would be necessary we would be happy to comply.

Reviewer 3 Report
Comments and Suggestions for Authors
This manuscript investigates the mechanical toughness and anatomical structure of cactus skin in four species to identify principles for bioinspired envelope design in technical applications.The topic is aligned with current interests in biomimetic design and resilient materials for extreme environments. The writing is generally clear. It may be recommended for publication after addressing the following concerns.
(a) Suggest expanding the literature review to include more biomechanical studies of plant cuticles and hypodermal structures under abiotic stress.
(b) Statistical relationships, particularly those with low R² values (e.g., Figure 5), should be more cautiously interpreted to avoid overstating significance.
(c) The quality of the graph is poor, including the axes, the font size and clarity of the labels.
Author Response
- Comment 1: Suggest expanding the literature review to include more biomechanical studies of plant cuticles and hypodermal structures under abiotic stress.
- Response 1: Thank you, we have added some references and new paragraphs on plant cuticle and hypodermal structure e.g. and aligned some discussion with points raised by referee 1 concerning the “uniqueness” and significance of the hypodermis in Cactaceae. Lines 107-133 (in green).
- Comment 2: Statistical relationships, particularly those with low R² values (e.g., Figure 5), should be more cautiously interpreted to avoid overstating significance.
- Response 2: Yes indeed we agree. Because the outer envelopes of these plants are what we might call “multi functional” it is perhaps not surprising that R² values might be low even though the slopes are significant. We have added some words to the existing text (In green, follows below) to emphasize this especially for the results in Figure 5 where toughness is explored along different ages of the stem.
"A plot of fracture toughness at variable distances from the apex for each measured segment showed no clear overall tendance when all four species were analysed together as shown by a very low overall R² value (Figure 5Aa). When analysed as distinct species, three out of the four species showed significant slopes but weakly supported R² values for increases in toughness with increasing distance from the apex. In these cases, stem toughness explained only approximately 25% of the variation for P. arrabidae (R2 = 0.282; p < 0,01); P. ulei (R2 = 0.2619; p < 0,01) and S. setaceus (R2 = 0,2303; p < 0,01). However, the trend seen in the creeping-upright species C. fernambucensis, presented only an extremely low R2 value, with no statistical significance (R2 = 0,0002; p > 0,05). Lines 342-344, 347-348 (in green).
Furthermore we have included a very brief potential explanation for the low R² values mentioning the “multi-functionality” of the envelope in plants, as follows:
“…these trends showed only weak or non significant R² values suggesting that other factors influence skin structure and toughness. This is perhaps not surprising because of the high multi-functionality of the outer envelopes of these plants with functions such as gas exchange, water conservation and permeability to light…” Lines 455-458.
- Comment 3: The quality of the graph is poor, including the axes, the font size and clarity of the labels.
- Response 3: Thank you for pointing this out, the graphics in figure 5 have been improved: font size and clarity of the legends.

Round 2
Reviewer 2 Report
Comments and Suggestions for Authors.